# Randomized phase II study of daily versus alternate-day administrations of S-1 for the elderly patients with completely resected pathological stage IA (tumor diameter > 2 cm) —IIIA of non-small cell lung cancer: Setouchi Lung Cancer Group Study 1201

**Hiromasa Yamamoto**[1]*, **Junichi Soh**[2], **Norihito Okumura**[3], **Hiroyuki Suzuki**[4], **Masao Nakata**[5], **Toshiya Fujiwara**[6], **Kenichi Gemba**[7], **Isao Sano**[8], **Takuji Fujinaga**[9], **Masafumi Kataoka**[10], **Yasuhiro Terazaki**[11], **Nobukazu Fujimoto**[12], **Kazuhiko Kataoka**[13], **Shinji Kosaka**[14], **Motohiro Yamashita**[15], **Hidetoshi Inokawa**[16], **Masaaki Inoue**[17], **Hiroshige Nakamura**[18], **Yoshinori Yamashita**[19], **Katsuyuki Hotta**[20], **Hiroshige Yoshioka**[21], **Satoshi Morita**[22], **Keitaro Matsuo**[23,24], **Junichi Sakamoto**[25], **Hiroshi Date**[26], **Shinichi Toyooka**[1]

1 Department of Thoracic Surgery, Okayama University Hospital, Okayama, Japan, 2 Department of Surgery, Division of Thoracic Surgery, Kindai University Faculty of Medicine, Osaka-Sayama, Japan, 3 Department of Thoracic Surgery, Kurashiki Central Hospital, Kurashiki, Japan, 4 Department of Chest Surgery, Fukushima Medical University Hospital, Fukushima, Japan, 5 Department of General Thoracic Surgery, Kawasaki Medical School Hospital, Kurashiki, Japan, 6 Department of Thoracic Surgery, Hiroshima City Hiroshima Citizens Hospital, Hiroshima, Japan, 7 Department of Respiratory Medicine, Chugoku Central Hospital, Fukuyama, Japan, 8 Department of Respiratory Surgery, Japanese Red Cross Nagasaki Genbaku Hospital, Nagasaki, Japan, 9 Department of General Thoracic Surgery, National Hospital Organization Nagara Medical Center, Gifu, Japan, 10 Department of Surgery and Respiratory Center, Okayama Saiseikai General Hospital, Okayama, Japan, 11 Department of Respiratory Surgery, Saga-Ken Medical Centre Koseikan, Saga, Japan, 12 Department of Medical Oncology and Respiratory Medicine, Okayama Rosai Hospital, Okayama, Japan, 13 Department of Thoracic Surgery, National Hospital Organization Iwakuni Clinical Center, Iwakuni, Japan, 14 Department of Thoracic Surgery, Shimane Prefectural Central Hospital, Izumo, Japan, 15 Department of Thoracic Surgery, National Hospital Organization Shikoku Cancer Center, Matsuyama, Japan, 16 Department of Thoracic Surgery, National Hospital Organization Yamaguchi-Ube Medical Center, Ube, Japan, 17 Department of Chest Surgery, Shimonoseki City Hospital, Shimonoseki, Japan, 18 Division of General Thoracic Surgery, Tottori University Hospital, Yonago, Japan, 19 Department of Thoracic Surgery, National Hospital Organization Kure Medical Center and Chugoku Cancer Center, Kure, Japan, 20 Center for Innovative Clinical Medicine, Okayama University Hospital, Okayama, Japan, 21 Department of Thoracic Oncology, Kansai Medical University Hospital, Hirakata, Japan, 22 Department of Biomedical Statistics and Bioinformatics, Kyoto University Graduate School of Medicine, Kyoto, Japan, 23 Division of Cancer Epidemiology and Prevention, Aichi Cancer Center Research Institute, Nagoya, Japan, 24 Department of Preventive Medicine, Kyushu University Faculty of Medical Sciences, Fukuoka, Japan, 25 Tokai Central Hospital, Kakamigahara, Japan, 26 Department of Thoracic Surgery, Kyoto University Hospital, Kyoto, Japan

* h.yamamoto@md.okayama-u.ac.jp

## Abstract

### Background

It is shown that the postoperative adjuvant chemotherapy for non-small cell lung cancer (NSCLC) was associated with survival benefit in an elderly population. We aimed to analyze the feasibility and efficacy of alternate-day S-1, an oral fluoropyrimidine, for adjuvant

**Data Availability Statement:** All relevant data are within the paper and its Supporting Information files.

**Funding:** This work was supported by a non-profit organization Epidemiological & Clinical Research Information Network (ECRIN), Kyoto, Japan. The funder had no role in study design, data analysis, decision to publish, or preparation of the manuscript, except for data collection and management.

**Competing interests:** HY1 received honoraria from Taiho Pharmaceutical outside the work. JS1 received honoraria from Johnson and Johnson and Intuitive outside the work. HN received grants from Taiho Pharmaceutical outside the work. KH received honoraria from Pfizer, AstraZeneca, Chugai Pharmaceutical, Eli Lilly, Takeda Pharmaceutical, MSD, BMS, Ono Pharmaceutical, Taiho Pharmaceutical and Boehringer-Ingelheim outside the work, and grants from MSD, AstraZeneca, Chugai Pharmaceutical, Eli Lilly, BMS and Abbvie outside the work. HY2 received consulting fees from Delta-Fly Pharma and honoraria for lectures from AstraZeneca, Boehringer-Ingelheim, Chugai Pharmaceutical, Eli Lilly, Taiho Pharmaceutical, Ono Pharmaceutical, MSD, Novartis, BMS, Pfizer, Daiichi-Sankyo, Kyowa Kirin, Takeda Pharmaceutical, Nippon Kayaku and Otsuka Pharmaceutical outside the work. SM received honoraria for lectures from Taiho Pharmaceutical outside the work. HD received grants from Taiho Pharmaceutical outside the work. ST received grants from Chugai Pharmaceutical, Taiho Pharmaceutical, Eli Lilly, PFDeNA, MSD, AstraZeneca and a non-profit organization West Japan Oncology Group (WJOG) (supported by AstraZeneca for the operating expense of the clinical trial) outside the work. All other authors declared no conflicts of interest regarding this study. This does not alter our adherence to PLOS ONE policies on sharing data and materials.

**Abbreviations:** NSCLC, non-small cell lung cancer; RDI, relative dose intensity; CI, confidence interval; SD, standard deviation; UICC, Union Internationale Contre le Cancer; EGFR, epidermal growth factor receptor; RFS, recurrence-free survival; OS, overall survival.

chemotherapy in elderly patients with completely resected pathological stage IA (tumor diameter > 2 cm) to IIIA (UICC TNM Classification of Malignant Tumours, 7th edition) NSCLC.

## Methods

Elderly patients were randomly assigned to receive adjuvant chemotherapy for one year consisting of either alternate-day oral administration of S-1 (80 mg/m$^2$/day) for 4 days a week (Arm A) or a daily oral administration of S-1 (80 mg/m$^2$/day) for 14 consecutive days followed by 7-day rest (Arm B). The primary endpoint was feasibility (treatment completion rate), which was defined as the proportion of patients who completed the allocated intervention for 6 months with a relative dose intensity (RDI) of 70% or more.

## Results

We enrolled 101 patients in which 97 patients received S-1 treatment. The treatment completion rate at 6 months was 69.4% in Arm A and 64.6% in Arm B ($p = 0.67$). Treatment completion rate in Arm B tended to be lower compared to Arm A, as the treatment period becomes longer (at 9 and 12 months). RDI of S-1 at 12 months and completion of S-1 administration without dose reduction or postponement at 12 months was significantly better in Arm A than in Arm B ($p = 0.026$ and $p < 0.001$, respectively). Among adverse events, anorexia, skin symptoms and lacrimation of any grade were significantly more frequent in Arm B compared with Arm A ($p = 0.0036$, 0.023 and 0.031, respectively). The 5-year recurrence-free survival rates were 56.9% and 65.7% for Arm A and B, respectively ($p = 0.22$). The 5-year overall survival rates were 68.6% and 82.0% for Arm A and B, respectively ($p = 0.11$).

## Conclusion

Although several adverse effects were less frequent in Arm A, both alternate-day and daily oral administrations of S-1 were demonstrated to be feasible in elderly patients with completely resected NSCLC.

## Trial registration

Unique ID issued by UMIN: UMIN000007819 (Date of registration: Apr 25, 2012) https://center6.umin.ac.jp/cgi-open-bin/ctr_e/ctr_view.cgi?recptno=R000009128. Trial ID issued by jRCT: jRCTs061180089 (Date of registration: Mar 22, 2019, for a shift toward a "specified clinical trial" based on Clinical Trials Act in Japan) https://jrct.niph.go.jp/en-latest-detail/jRCTs061180089.

## Introduction

Surgical resection is considered to be the best curative treatment for early-stage non-small cell lung cancer (NSCLC) [1–3]. Despite the complete resection, however, disease relapse is often recognized. Even in the node-negative cases with a tumor diameter ≤ 3 cm, 15% of the patients develop distant recurrences [4].

In Japan, postoperative adjuvant chemotherapy with the oral drug tegafur/uracil (UFT) was reported to significantly improve survival in NSCLC patients, especially in adenocarcinoma

[5]. After the reanalysis to evaluate the effectiveness of UFT regarding the tumor size, UFT was shown to significantly improve survival in stage I NSCLC patients with a tumor diameter of 2 to 3 cm [6]. Furthermore, a recent randomized phase III trial conducted by our group demonstrated that no survival difference was found between UFT versus paclitaxel plus carboplatin as adjuvant chemotherapy for completely resected stage IB to IIIA NSCLC [7].

On the other hand, the postoperative adjuvant chemotherapy, especially platinum containing regimen, sometimes causes treatment-related death even in good risk patients. Thus, it is not always recommended for elderly patients who often had some complicating diseases and may have potentially impaired health condition. Meanwhile, there is a worldwide-accepted evidence of a population shift toward older ages, which favors an increased risk of developing lung cancer that are more common at a more advanced age [8]. The present aging society demands to develop adjuvant chemotherapy for elderly NSCLC patients for disease cure.

Although NSCLC represents a common health issue in the elderly population, it is difficult to provide evidence-based clinical recommendations for this population, because of the lack of large, well-conducted prospective trials. EORTC and International Society of Geriatric Oncology (SIOG) showed the recommendation for the treatment of NSCLC in an elderly population. They described that postoperative adjuvant chemotherapy is associated with survival benefit in the elderly and therefore should not be denied to patients based on age. Treatment decisions should consider the estimated absolute benefit, life expectancy, treatment tolerance, presence of comorbidities and patient preferences, although less information is available regarding the real benefit and tolerability of the regimens for patients aged > 80 years and the risk versus benefit has not been studied adequately [9].

A review regarding the management of elderly patients with NSCLC suggested the possibility for elderly patients to replace cisplatin to carboplatin for the better tolerability, describing the use of adjuvant chemotherapy in fit elderly patients, although data are insufficient to draw conclusions in patients aged > 75 years [10].

S-1 (Taiho Pharmaceutical Co., Ltd, Tokyo, Japan) is an oral fluoropyrimidine agent consisting of tegafur (a prodrug of 5-fluorouracil [5-FU]), gimeracil (an inhibitor of dihydropyrimidine dehydrogenase, which degrades fluorouracil), and oteracil (an inhibitor of the phosphorylation of fluorouracil in the gastrointestinal tract, resulting in reduction of the gastrointestinal toxic effects of fluorouracil) [11]. The efficacy and safety of S-1 was demonstrated in patients receiving initial chemotherapy for unresectable, advanced NSCLC, with the regimen of 4-week daily administration of S-1 followed by a 2-week rest [12]. In advanced head and neck cancer, the regimen of 2-week administration of S-1 followed by 1-week rest was compared to the existing regimen of 4-week administration of S-1 followed by 2-week rest, and 2-week administration of S-1 followed by 1-week rest seemed to be more feasible [13]. Multiple phase II single-arm Japanese studies have also demonstrated encouraging efficacy of S-1 as monotherapy in treatment-naïve elderly patients with advanced NSCLC, a population that frequently cannot tolerate platinum-doublet chemotherapy [14].

As postoperative adjuvant chemotherapy for stage I (tumor diameter greater than 2 cm) disease, the Japan Clinical Oncology Group conducted a randomized phase III study to evaluate S-1 compared with UFT (JCOG0707), with the regimen of 2-week daily administration of S-1 followed by a 1-week rest, in which S-1 was neither superior nor inferior to UFT [15]. As a reported randomized phase II study, S-1 monotherapy might be preferable to cisplatin plus S-1 in pathological stage (pStage) II-III disease, indicating a promising efficacy of adjuvant S-1 monotherapy, however, the completion rate of two-week daily S-1 administration followed by one-week rest for about 1 year showed only 52.6% [16]. Thus, it is important to develop more feasible administration schedule of S-1 in the adjuvant setting.

 

To improve the feasibility of S-1, alternate-day administration of S-1 was suggested. In advanced gastric cancer, alternate-day administration (every other day) of S-1 demonstrated the reduction of adverse effects, simultaneously ensured effective blood levels and provided sufficient clinical effects compared with daily administration [17]. In a randomized control trial, alternate-day S-1 administration as postoperative adjuvant therapy showed reducing adverse effect with maintaining the drug efficacy compared to daily S-1 administration in gastric cancer patients [18]. This fact suggests that alternate-day S-1 administration may be more feasible adjuvant chemotherapy with the same efficacy in NSCLC patients when compared with daily S-1 administration, suggesting that this regimen may be suitable especially for elderly patients.

Because elderly patients frequently cannot tolerate platinum-doublet chemotherapy, and S-1 monotherapy is shown to be effective to advanced NSCLC in elderly patients, it is reasonable to set S-1 monotherapy as postoperative adjuvant chemotherapy in elderly patients with completely resected NSCLC including pStage II-III.

To our knowledge, however, there are no reports of prospective studies using an alternate-day S-1 administration of adjuvant chemotherapy for elderly patients with completely resected NSCLC. Considering these, we designed a multicenter, randomized phase II trial comparing the feasibility and safety of alternate-day administration versus daily administration for 2 weeks followed by a 1-week rest as adjuvant S-1 monotherapy in elderly patients with completely resected stage IA (T1bN0M0)—IIIA NSCLC.

## Patients and methods

### Patients' selection

Patients who met all the following eligibility criteria and none of the exclusion criteria were enrolled in this study. The eligibility criteria were as follows: (i) pathologically proven NSCLC with complete resection, (ii) pStage IA (T1bN0M0)-IIIA (according to the Union Internationale Contre le Cancer [UICC] TNM Classification of Malignant Tumours, 7th edition) [19, 20], (iii) lobectomy or larger lung resection with complete lymph node dissection (ND2a and more extensive in principle), (iv) patients who were able to begin the protocol treatment within 8 weeks after surgical resection, (v) no prior chemotherapy or radiotherapy, (vi) age of 75 years or older at the time of the enrollment, (vii) an Eastern Cooperative Oncology Group performance status 0 or 1, (viii) adequate organ function [leukocytes $\geq$ 3,000/μl and $\leq$ 12,000/μl, platelets $\geq$ 100,000/μl, hemoglobin $\geq$ 9.0 g/dl, total bilirubin $\leq$ 1.5 mg/dl, aspartate aminotransferase and alanine aminotransferase each $\leq$ 100 IU/l, creatinine clearance $\geq$ 40 ml/min, $PaO_2 \geq$ 60 mmHg or SpO2 $\geq$ 90%], and (ix) written informed consent prior to enrollment. The details of exclusion criteria are listed in S1 Table.

### Treatment and follow-up

The principal investigator/participating investigator confirmed that the eligible patients met all the eligibility criteria and did not violate the exclusion criteria, filled out the "Case Registration Form", and faxed the form to the research secretariat (Department of Thoracic Surgery, Okayama University Hospital, Okayama, Japan). After the research secretariat confirmed eligibility, it faxed back a notification of case registration results to the investigator. Upon enrollment, each patient was randomly assigned to a treatment by Dr. Keitaro Matsuo, Department of Preventive Medicine, Kyushu University Faculty of Medical Sciences, Fukuoka, Japan. Random assignment was performed using a minimization method with the following adjustment factors: pStage [stage IA (T1bN0M0)/IB or IIA/IIB or IIIA], histology (squamous cell carcinoma or non-squamous cell carcinoma), epidermal growth factor (*EGFR*) mutational

status (positive or negative) and institution. Patients received the allocated intervention in each participating institution. Patients received S-1 twice per day in the morning and in the evening, by equally dividing the amount of administration per day into two. The dose of S-1 was 80 mg/body/day when the body surface area was $< 1.25$ m$^2$, 100 mg/body/day for 1.25– 1.50 m$^2$, and 120 mg/body/day for $> 1.50$ m$^2$. Patients were randomly assigned to one of the two regimens for S-1; S-1 was administered on Monday, Wednesday, Friday and Sunday of every week (alternate-day administration, Arm A) or daily for 2 weeks followed by a 1-week rest (daily administration, Arm B). These cycles were repeated every week (Arm A) or every 3 weeks (Arm B) for 1 year after the start of oral administration. For Arm B, S-1 was adminis- tered until day 14 of the final cycle (Cycle #18). Planned total dose of S-1 is 80, 100, 120 mg/ day (initial dose of S-1) x 4 days/week x 52 weeks (208 days) for Arm A, and 80, 100, 120 mg/ day (initial dose of S-1) x 18 cycles x 14 days (252 days) for Arm B (S1 Fig). The details of the criteria for discontinuation and restart of S-1 administration, the criteria and manner of dose reduction, and the criteria for cessation of the treatment protocol are provided in S2–S5 Tables, respectively.

As for baseline evaluations, medical history, smoking history, physical examination, opera- tion date, p-TNM status, tumor histology (squamous cell carcinoma or non-squamous cell car- cinoma), comorbidity, and laboratory analyses were included. The details of the follow-up assessments are provided in S6 Table. Toxicity was graded according to the Common Termi- nology Criteria for Adverse Events, version 4.0.

## Study design and statistical analysis

The study was designed as a multicenter randomized phase II study, conducted in accordance with the Declaration of Helsinki, and registered with the University hospital Medical Informa- tion Network (UMIN) Clinical Trials Registry (UMIN-CTR) (Unique ID issued by UMIN: UMIN000007819) (https://center6.umin.ac.jp/cgi-open-bin/ctr_e/ctr_view.cgi?recptno= R000009128). The study protocol was approved by the institutional review board of each par- ticipating institution. After that, this study moved toward a "specified clinical trial" based on Clinical Trials Act in Japan and was registered with the Japan Registry of Clinical Trials (jRCT) (Trial ID issued by jRCT: jRCTs061180089) (https://jrct.niph.go.jp/en-latest-detail/ jRCTs061180089). It was approved by Okayama University Certified Review Board, followed by the confirmation of each participating institution. All the study data were managed by the SLCG1201 data center at a non-profit organization, the Epidemiological and Clinical Research Information Network (ECRIN), Kyoto, Japan, which anonymized each participant by assign- ment of a new number. The primary endpoint of this study was feasibility (treatment comple- tion rate), which was defined as the proportion of patients who completed the allocated intervention for 6 months with 70% or more of relative dose intensity (RDI). RDI was defined as the rate between the actual total administered dose (based on the self-reported dose of the patients who took medication of S-1) and the planned total administered dose. Patients who discontinued the treatment protocol because of tumor recurrence or other complications unrelated to S-1 were treated as censored cases. Treatment completion rates under the condi- tions described above at 9 months and at 12 months after the initiation of the treatment proto- col were also calculated. The continuation rate of S-1 administration was also evaluated.

This study was designed according to a randomized phase II selection design [21]. We assumed that the threshold 6-month treatment completion rate for the current protocol in both groups was 40%. The decision criteria for the primary endpoint were as follows: 1) If the 6-month treatment completion rate is 40% or less in both groups, the protocol treatments for both groups is considered not promising for postoperative adjuvant chemotherapy in the

elderly patients with completely resected stage IA(T1bN0M0)-IIIA NSCLC. 2) If the 6-month treatment completion rate of one group is more than 40% and exceeds that of another group by at least 15%, the regimen for the group in which the completion rate was higher is considered promising and selected for further phase III study. 3) If the 6-month treatment completion rate of one group is more than 40% and exceeds that of another group by 15% or less, the comparison between the two groups should not be performed with the treatment completion rate alone, but rather with toxicity, quality of life, convenience, cost, treatment completion rate after 6 months, recurrence-free survival (RFS), and overall survival (OS) to determine which is more suitable as S-1 therapy in future Phase III trials. According to the design for assuring 90% probability for selecting the best study arm, if the true expected completion rate exceeded that of another arm by at least 15%, we estimated that the required number of patients would be 37 patients for each arm. Finally, the sample size was set to 100 considering the potential for patient drop-out because of ineligibility.

The secondary endpoints were toxicity, RFS, and OS. A final analysis of survival time was done 5 years after the last enrollment.

Significant differences between the categorized groups were compared using the Chi-square test or Fisher's exact test. For the analyses of continuous values, t-test or Mann-Whitney U test was used. Univariate analysis of the continuation rate of S-1 administration, OS and RFS was performed using the Kaplan-Meier method with log-rank testing as intent-to-treat analyses. We defined $p < 0.05$ as the threshold for statistical significance. All statistical analyses in this study were performed using the EZR (version 1.55, Saitama Medical Center, Jichi Medical University, Saitama, Japan), which is a graphical user interface for R (version 4.1.2, The R Foundation for Statistical Computing, Vienna, Austria) [22]. More precisely, the software is a modified version of R Commander (version 2.7–2), designed to add statistical functions frequently used in biostatistics.

### Ethics approval and consent to participate

This study was approved by Ethics Committee, Okayama University Graduate School of Medicine, Dentistry and Pharmaceutical Sciences and Okayama University Hospital, Okayama, Japan (approval number: Rin1967), which was a main research institution in this multicenter prospective phase II study, and then it was also approved by the institutional review board of each participating institution. Written informed consent was taken from each patient. This study moved toward a "specified clinical trial" based on Clinical Trials Act in Japan and was approved by Okayama University Certified Review Board (approval number: CRB18-011), followed by the confirmation of each participating institution.

## Results

### Patient characteristics

We enrolled 101 patients in this trial from 19 institutions in Japan from May 2012 to April 2016. The number of enrolled and candidate cases in each participating institution is shown in S7 Table. We were not able to obtain the number of candidate cases from 3 out of 19 participating institutions. Among 16 institutions that had the data of candidate cases, 94 cases were enrolled from 872 cases. Three patients refused the protocol treatment after the agreement of the study participation, and one patient did not receive the protocol treatment because of the poor general condition. Finally, 97 patients received the allocated intervention (49 in Arm A and 48 in Arm B) (Fig 1). The baseline characteristics of the enrolled patients are summarized in Table 1. Seventy-four patients (73.3%) were men, and the median age was 77 years old. Seventy-three (72.3%) patients had non-squamous cell carcinoma histology and 57 (56.4%)

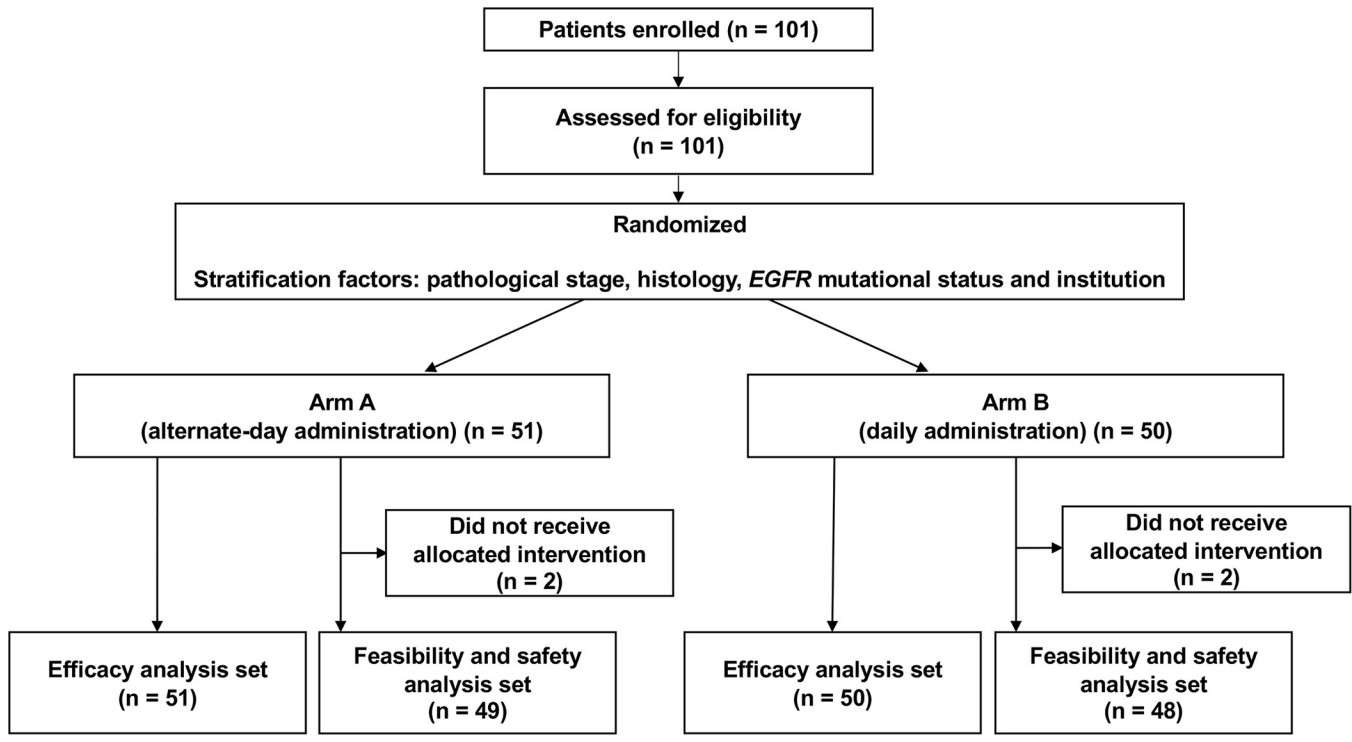

**Fig 1. Consolidated Standards of Reporting Trials (CONSORT) diagram of this study.**

patients were pStage IA (T1bN0M0) / IB. *EGFR* mutation was detected in 31 (30.7%) patients. There were no serious adverse event resulting in discontinuation the entire study.

## Feasibility

Treatment discontinuation rates at 12 months were 30.6% and 50.0% in Arm A and B, respectively ($p = 0.064$). Treatment was discontinued because of adverse events or toxicity in 8 (16.3%) and 12 (25.0%) patients in Arm A and B, respectively ($p = 0.33$). Median RDI of S-1 at 12 months was 83.4% for Arm A and 68.6% for Arm B, respectively ($p = 0.026$). Median actual total administered dose of S-1 was 17340 mg for Arm A and 17640 md for Arm B, respectively ($p = 0.82$) (Table 2).

Treatment completion rate, which is the proportion of the patients who completed the administration of S-1 during the period (6 months, 9 months and 12 months) with 70% or more of RDI, was as follows: 69.4% for Arm A and 64.6% for Arm B at 6 months, 67.3% for Arm A and 56.3% for Arm B at 9 months, and 63.3% for Arm A and 45.8% for Arm B at 12 months. During the 1-year treatment course, 51.0% of the patients in Arm A and 16.7% of the patients in Arm B completed S-1 administration without a dose reduction or postponement ($p < 0.001$) (Table 3).

Continuation rate of S-1 administration at 100 days, 150 days and 200 days was 87.2%, 82.3% and 82.3% in Arm A and 84.2%, 79.1% and 70.9% in Arm B, respectively ($p = 0.29$) (Fig 2).

The treatment completion rate based on an RDI of 70% or more for 6 months in Arm A was greater than 40% and it did not exceed that of Arm B by more than 15%, indicating that both Arm A and Arm B were feasible. To determine which is more suitable as S-1 therapy in future Phase III trials, the comparison between the two groups should not be performed with the treatment completion rate alone, but rather with toxicity, quality of life, convenience, cost, treatment completion rate after 6 months, RFS, and OS.

**Table 1. Characteristics of the elderly patients with completely resected p-stage IA-IIIA non-small cell lung cancer (n = 101).**

| Variables | Total (n = 101) | | Arm A (n = 51) | | Arm B (n = 50) | | p value |
|---|---|---|---|---|---|---|---|
| | n | % | n | % | n | % | |
| Age | | | | | | | 0.88 |
| Median (range) | 77 (75–87) | | 78 (75–84) | | 77 (75–87) | | |
| Sex | | | | | | | 0.063 |
| Male | 74 | 73.3 | 42 | 82.4 | 32 | 64.0 | |
| Female | 27 | 26.7 | 9 | 17.6 | 18 | 36.0 | |
| ECOG PS | | | | | | | 0.95 |
| 0 | 70 | 69.3 | 36 | 70.6 | 34 | 68.0 | |
| 1 | 31 | 30.7 | 15 | 29.4 | 16 | 32.0 | |
| Smoking status | | | | | | | 0.043 |
| Ever | 71 | 70.3 | 41 | 80.4 | 30 | 60.0 | |
| Never | 30 | 28.7 | 10 | 19.6 | 20 | 40.0 | |
| Histological subtypes | | | | | | | 0.78 |
| Sq | 28 | 27.7 | 13 | 25.5 | 15 | 30.0 | |
| Non-Sq | 73 | 72.3 | 38 | 74.5 | 35 | 70.0 | |
| pStage* | | | | | | | 1.00 |
| IA (T1bN0M0) / IB | 57 | 56.4 | 29 | 56.9 | 28 | 56.0 | |
| IIA / IIB | 26 | 25.7 | 13 | 25.5 | 13 | 26.0 | |
| IIIA | 18 | 17.8 | 9 | 17.6 | 9 | 18.0 | |
| *EGFR* mutational status** | | | | | | | 1.00 |
| Positive | 31 | 30.7 | 16 | 31.4 | 15 | 30.0 | |
| Negative or unknown | 70 | 69.3 | 35 | 68.6 | 35 | 70.0 | |

Sq, squamous cell carcinoma

ECOG PS, Eastern Cooperative Oncology Group performance status

*EGFR*, epidermal growth factor receptor

Arm A: alternative-day administration regimen

Arm B: daily administration regimen

*UICC TNM Classification of Malignant Tumours, 7th edition

**EGFR mutational status was unknown in one case with squamous cell carcinoma in Arm A.

## Toxicity

A summary of the adverse events is shown in Table 4. Toxicities were generally well tolerated in both groups and there were no grade 4 adverse events or treatment-related deaths for any patient. The incidence of any adverse events of any grade was 98.0% and 97.9% in Arm A and B, respectively. As for moderate or severe adverse events (grade 2 or grade 3), the incidence was 59.2% and 70.8% in Arm A and B, respectively ($p = 0.29$). The main adverse events were hematological, gastrointestinal, and cutaneous symptoms. Among the adverse events, anorexia, skin symptoms, and lacrimation were significantly more frequent in Arm B compared with Arm A (54.2% vs. 24.5%; $p = 0.0036$, 37.5% vs. 16.3%; $p = 0.023$ and 25.0% vs. 8.2%; $p = 0.031$, respectively).

## Survival

In Arm A, 21 patients died in the study period consisting of 9 patients for current lung cancer, 3 for other malignancies and 9 for other diseases or accidents. In Arm B, 14 patients died consisting of 8 patients for current lung cancer, 1 for other malignancies and 5 for other diseases or accidents (S8 Table).

**Table 2. Treatment discontinuation, relative dose intensity at 12 months and actual total administered dose of S-1 (n = 97).**

| | Arm A (n = 49) | Arm B (B = 48) | *p* value |
|---|---|---|---|
| Treatment discontinuation | 15 (30.6%) | 24 (50.0%) | 0.064 |
| Reason for discontinuation | | | |
| Toxicity or adverse events | 8 (16.3%) | 12 (25.0%) | 0.33 |
| Recurrence | 4 (8.2%) | 2 (4.2%) | 0.68 |
| 2nd malignancy | 0 (0%) | 0 (0%) | 1.000 |
| Patient refusal | 1 (2.0%) | 4 (8.3%) | 0.20 |
| Investigator decision | 0 (0%) | 2 (4.2%) | 0.24 |
| Others | 2 (4.1%) | 4 (8.3%) | 0.44 |
| Relative dose intensity of S-1 at 12 months | | | 0.026 |
| Median | 83.4% | 68.6% | |
| Range | 0.481–102.4% | 2.033–100% | |
| Actual total administered dose of S-1 (mg) | | | 0.82 |
| Median | 17340 | 17640 | |
| Range | 100–25320 | 500–30240 | |

The median follow-up time for all enrolled patients (n = 101) was 71.9 months (range, 6.2 to 103.9). Patients who received the protocol treatment (n = 97) and all enrolled patients (n = 101) were monitored until death or for at least 5 years from the registration. Survival analyses were performed based on an intention to treat. The 5-year RFS rate for all patients (n = 101) was 56.9% and 65.7% for Arm A and B, respectively (*p* = 0.22) (Fig 3A). The 5-year OS rate for all patients (n = 101) was 68.6% and 82.0% for Arm A and B, respectively (*p* = 0.11) (Fig 3B). Regarding disease-specific (lung cancer-specific) survival, the 5-year survival rate for all patients (n = 101) was 82.8% and 89.5% for Arm A and B, respectively (*p* = 0.54) (S2 Fig).

## Subset analysis of survival

Regarding the impact of the treatment completion rate of S-1 administration at 6 months, it seems that RFS and OS were better in the cases with the completion of S-1 administration (S3A, S3C Fig). However, there were no significant differences of RFS or OS if excluding the recurrent cases within 6 months (S3B, S3D Fig, 6 cases were excluded). For the group with the completion of S-1 administration at 6 months, there were no significant differences of RFS or OS (S3E, S3F Fig). For the group without the completion of S-1 administration at 6 months, RFS and OS tended to be worse in Arm A compared to Arm B. (S3G–S3J Fig). We also

**Table 3. Treatment completion rate (fraction of the cases with the continuation of the administration during the period and relative dose intensity ≥ 70%) (n = 97).**

| | Arm A (n = 49) | Arm B (n = 48) | *p* value |
|---|---|---|---|
| 6 months | 34 (69.4%) | 31 (64.6%) | 0.67 |
| among the cases excluding the discontinuation by recurrence | 34/45 (75.6%) | 31/46 (67.4%) | 0.49 |
| 9 months | 33 (67.3%) | 27 (56.3%) | 0.30 |
| among the cases excluding the discontinuation by recurrence | 33/45 (73.3%) | 27/46 (58.7%) | 0.19 |
| 12 months | 31 (63.3%) | 22 (45.8%) | 0.10 |
| among the cases excluding the discontinuation by recurrence | 31/45 (68.9%) | 22/46 (47.8%) | 0.056 |
| Completion of S-1 administration without dose reduction or postponement at 12 months | 25 (51.0%) | 8 (16.7%) | < 0.001 |

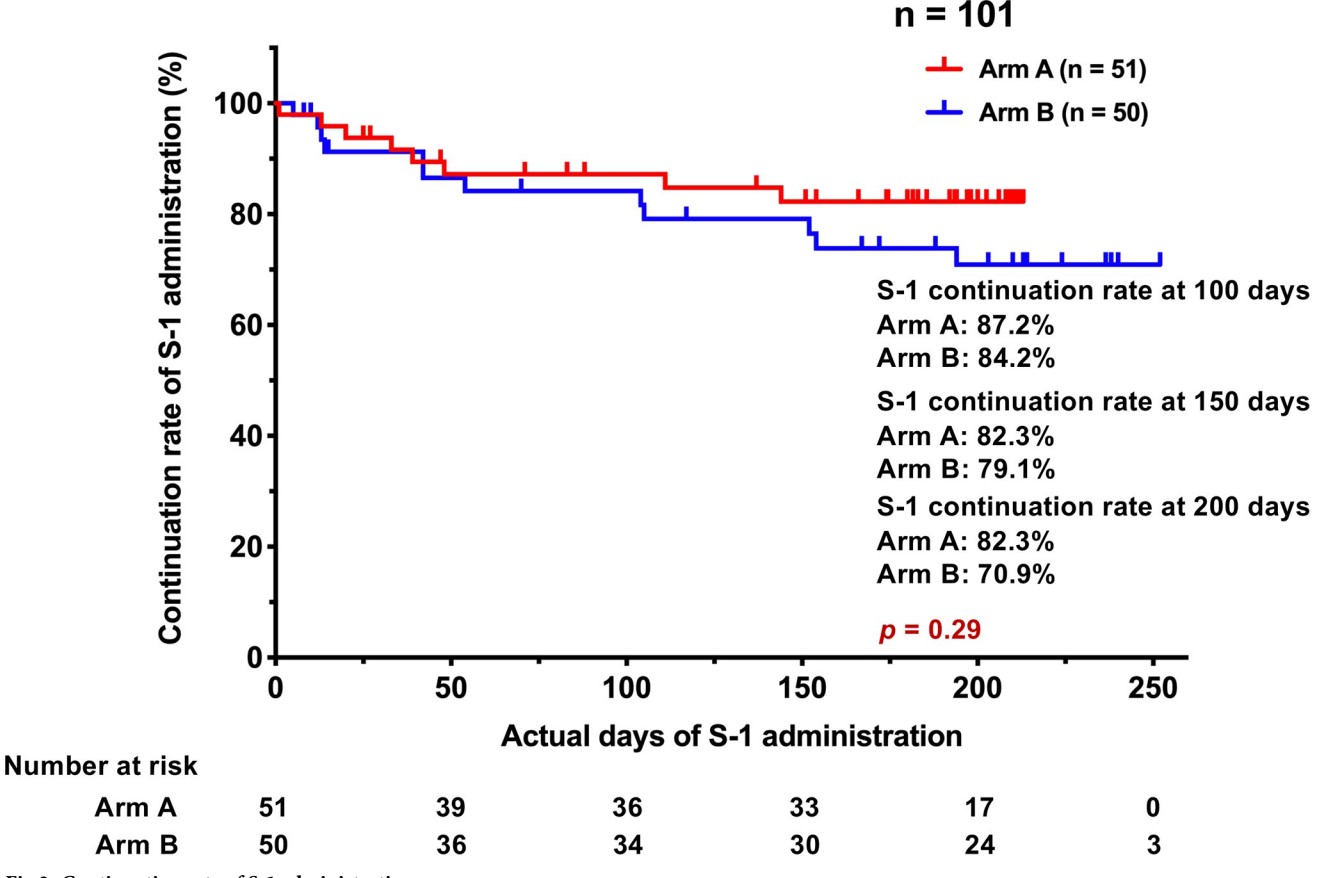

**Fig 2. Continuation rate of S-1 administration.**

evaluated the disease-specific survival for the cases with or without the completion of S-1 administration at 6 months. There were no significant differences of disease-specific survival in both arms (S3K–S3M Fig). We focused on the actual administered dose of S-1 in patients with or without treatment completion at 6 months. However, there were no significant differences of actual administered dose of S-1 between Arm A and Arm B in patients with or without treatment completion at 6 months (S9 Table).

We also analyzed RFS, and OS stratified by pStage (IA/IB, IIA/IIB, and IIIA). Both RFS and OS were better in earlier pStage (S4A, S4B Fig). As for RFS, there were no significant differences between Arm A and Arm B in the patients with pStage IA/IB (S4C Fig) and with pStage IB (S4D Fig). As for OS, Arm A was worse in the patients with pStage IA/IB (S4E Fig), especially with pStage IB (S4F Fig). There were no significant differences of RFS or OS between Arm A and Arm B in the patients with pStage IIA/IIB or IIIA (S4G–S4J Fig). Indeed, sex and smoking status were not included in the adjustment factors of the randomization and the frequency of male and ever smokers was more in Arm A (Table 1). Thus, we checked the distribution of them per pStage. As shown in S10 and S11 Tables, both male and ever smokers in pStage IB are significantly more frequent in Arm A than in Arm B ($p$ = 0.022 and 0.0028, respectively) (S10 and S11 Tables).

Regarding *EGFR* mutational status, there was no significant difference in RFS (S5A Fig), whereas OS was better in the cases with *EGFR* mutation (S5B Fig). As for smoking status and sex, both RFS and OS were worse in the cases with ever smoking history or in male cases, respectively (S6A, S6B and S7A, S7B Figs).

**Table 4.** Adverse events (n = 97).

| Adverse events | Arm A (n = 49) | | | | | Arm B (n = 48) | | | | | p value | |
|---|---|---|---|---|---|---|---|---|---|---|---|---|
| | G1 | G2 | G3 | Any grade (%) | G2/G3 (%) | G1 | G2 | G3 | Any grade (%) | G2/G3 (%) | Any grade | G2/G3 |
| Any adverse events | 19 | 24 | 5 | 48 (98.0%) | 29 (59.2%) | 13 | 26 | 8 | 47 (97.9%) | 34 (70.8%) | 1.00 | 0.29 |
| Anemia | 15 | 4 | 0 | 19 (38.8%) | 4 (8.2%) | 19 | 5 | 1 | 25 (52.1%) | 6 (12.5%) | 0.22 | 0.52 |
| Leukopenia | 5 | 5 | 0 | 10 (20.4%) | 5 (10.2%) | 4 | 5 | 1 | 10 (20.8%) | 6 (12.5%) | 1.00 | 0.76 |
| Neutropenia | 3 | 0 | 1 | 4 (8.2%) | 1 (2.0%) | 3 | 1 | 1 | 5 (10.4%) | 2 (4.2%) | 0.74 | 0.62 |
| Thrombocytopenia | 12 | 2 | 0 | 14 (28.6%) | 2 (4.1%) | 17 | 3 | 0 | 20 (41.7%) | 3 (6.3%) | 0.21 | 0.68 |
| Elevation of AST | 5 | 0 | 2 | 7 (14.3%) | 2 (4.1%) | 9 | 0 | 0 | 9 (18.8%) | 0 (0%) | 0.60 | 0.50 |
| Elevation of ALT | 3 | 1 | 1 | 5 (10.2%) | 2 (4.1%) | 1 | 0 | 0 | 1 (2.1%) | 0 (0%) | 0.20 | 0.50 |
| Elevation of LDH | 11 | 0 | 0 | 11 (22.4%) | 0 (0%) | 13 | 0 | 0 | 13 (27.1%) | 0 (0%) | 0.64 | 1.00 |
| Elevation of bilirubin | 8 | 8 | 1 | 17 (34.7%) | 9 (18.4%) | 12 | 4 | 0 | 16 (33.3%) | 4 (8.3%) | 1.00 | 0.23 |
| Elevation of creatinine | 7 | 0 | 0 | 7 (14.3%) | 0 (0%) | 4 | 0 | 0 | 4 (8.3%) | 0 (0%) | 0.52 | 1.00 |
| Anorexia | 7 | 3 | 2 | 12 (24.5%) | 5 (10.2%) | 18 | 5 | 3 | 26 (54.2%) | 8 (16.7%) | 0.0036 | 0.39 |
| Nausea | 5 | 1 | 0 | 6 (12.2%) | 1 (2.0%) | 6 | 4 | 0 | 10 (20.8%) | 4 (8.3%) | 0.29 | 0.20 |
| Vomiting | 3 | 1 | 0 | 4 (8.2%) | 1 (2.0%) | 0 | 1 | 0 | 1 (2.1%) | 1 (2.1%) | 0.36 | 1.00 |
| Diarrhea | 8 | 2 | 0 | 10 (20.4%) | 2 (4.1%) | 4 | 2 | 0 | 6 (12.5%) | 2 (4.2%) | 0.41 | 1.00 |
| Stomatitis | 4 | 2 | 0 | 6 (12.2%) | 2 (4.1%) | 4 | 3 | 2 | 9 (18.8%) | 5 (10.4%) | 0.41 | 0.27 |
| Fatigue | 3 | 1 | 0 | 4 (8.2%) | 1 (2.0%) | 9 | 1 | 1 | 11 (22.9%) | 2 (4.2%) | 0.053 | 0.62 |
| Skin symptoms | 7 | 1 | 0 | 8 (16.3%) | 1 (2.0%) | 14 | 4 | 0 | 18 (37.5%) | 4 (8.3%) | 0.023 | 0.20 |
| Lacrimation | 3 | 1 | 0 | 4 (8.2%) | 1 (2.0%) | 4 | 7 | 1 | 12 (25.0%) | 8 (16.7%) | 0.031 | 0.016 |

## Discussion

The advantage of alternate-day S-1 administration instead of daily use of S-1 is to reduce adverse effect especially of the gastrointestinal tract while maintaining the anticancer effect [23]. The rationale of alternate-day S-1 administration from the viewpoint of the differences in cell cycles between cancer and normal cells has been reported [24]. Regarding the feasibility of daily S-1 administration schedule in resected NSCLC patients, we also conducted randomized feasibility trial to compare the compliance, namely drug discontinuation free survival rate, of two types of daily S-1 administration schedule for one year in patients with completely-resected stage IA (tumor diameter, 2 to 3 cm) NSCLC: 4-week administration followed by a 2-week rest period (Group A) and a 2-week administration followed by a 1-week rest period

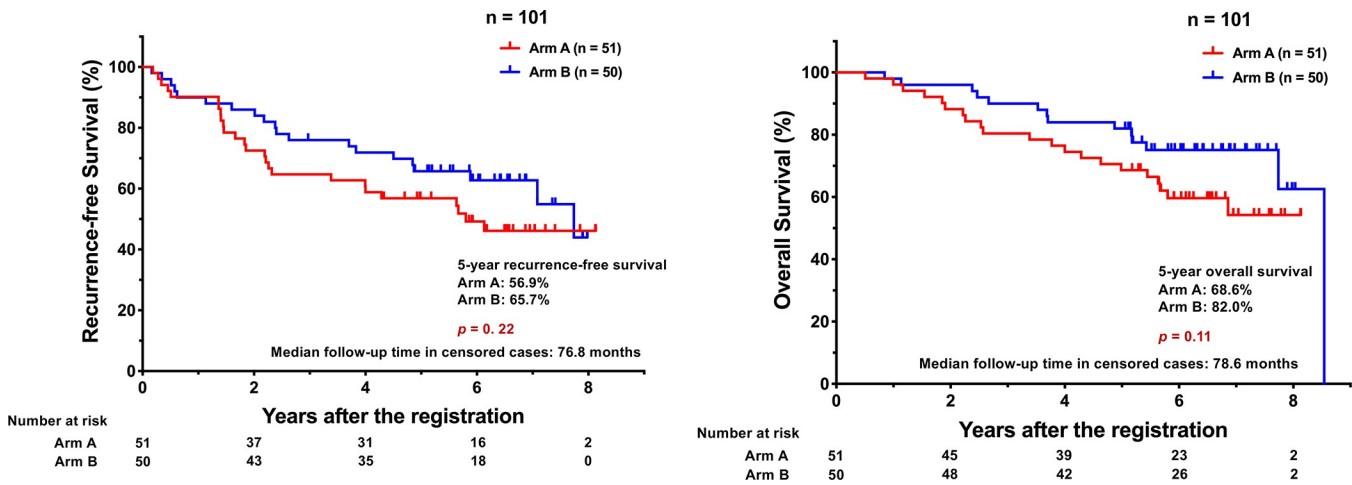

**Fig 3. (A)** Recurrence-free survival (RFS) of all the patients enrolled in this study. **(B)** Overall survival (OS) of all the patients enrolled in this study.

(Group B), which is the same as the Arm B in the current study (Setouchi Lung Cancer Group Study 0701, UMIN000006967) [25]. In that study, patients with grade 3/4 toxicities were significantly more frequent in Group A (40.5%) than in Group B (15.4%, $p$ = 0.021), indicating that the Group B schedule might be more tolerable. However, its drug discontinuation free survival rate, which is similar to the treatment completion rate at 1 year, was 52.7%, lower than 65% (hypothesis in that study [25] based on the previous report [13]), suggesting that the compliance is not satisfactory despite no treatment-related death and low rate of any severe adverse events. We also conducted a similar randomized feasibility trial of S-1 for NSCLC patients younger than 75 years [Setouchi Lung Cancer Group Study 1301 (SLCG1301), UMIN000011994, jRCTs061180082] [26]. In that study, the treatment completion rate based on RDI of 70% or more for 6 months was 84.4% in the group of alternate-day administration of S-1 (Group A) and 64.4% in the group of 2-week administration of S-1 followed by 1-week rest (Group B), and moderate or severe adverse events (grade 2 or grade 3) were significantly more frequent in Group B. In the current study, any grade of anorexia, skin symptoms, and lacrimation were significantly more frequent in Arm B compared with Arm A. Although it is similar for the group of alternate-day administration of S-1 to have less frequent adverse events, the advantage of alternate-day administration of S-1 in the toxicity was slightly minor in NSCLC patients with the age of 75 or older.

In the current study, both alternate-administration of S-1 (Arm A) and daily administration (Arm B) were feasible in terms of the treatment completion at 6 months, whereas there is a trend of unfavorable prognosis in Arm A. Especially, the trend of unfavorable prognosis is noted in patients without completion of S-1 administration at 6 months. However, there were no significant differences of actual administered dose of S-1 between Arm A and Arm B in patients without treatment completion at 6 months. The result of disease-specific survival showed no difference in both arms, suggesting the competing risks, such as noncancer deaths and secondary malignancies. As for patient characteristics, Arm A had male and ever smoker cases more frequently compared to Arm B, which was different from our similar study SLCG1301 [26]. Especially, both male and ever smoker cases with pStage IB were more frequent in Arm A than in Arm B. Male and ever smoking status are both reported to be poor prognostic factors in NSCLC [27–30]. As shown in S6 and S7 Figs, these factors were also shown to be poor prognostic factors in the current study. They may have affected the trend of unfavorable prognosis in Arm A. As shown in Table 2, RDI of S-1 at 12 months is significantly better in Arm A than in Arm B. As for the treatment completion rate, Table 3 shows that the completion rate at 12 months in Arm A (63.3%) is more than 15% higher than that in Arm B (45.8%). Table 3 also shows that completion of S-1 administration without dose reduction or postponement at 12 months was significantly better in Arm A than in Arm B. Regarding adverse events, anorexia, skin symptoms, and lacrimation were significantly more frequent in Arm B compared with Arm A as shown in Table 4. In clinical practice, which treatment arm to select for elderly patients with completely resected NSCLC depends on the physician's discretion in consideration of the results above.

It is important but difficult to select the appropriate patients after surgical resection for postoperative adjuvant chemotherapy. A recent study indicated that the detection of molecular residual disease (MRD) by circulating tumor DNA (ctDNA) profiling identified residual/recurrent disease earlier than standard-of-care radiologic imaging, and thus could facilitate personalized adjuvant treatment at early time points when disease burden is lowest [31]. If ctDNA technology improves its sensitivity, there is the possibility that postoperative adjuvant chemotherapy could be withheld from patients without residual disease [32]. Such technologies should contribute to the selection of the appropriate elderly patients for adjuvant chemotherapy.

The current study has some limitation. Planned total dose of S-1 was different. Total planned dose of S-1 in Arm A is about 82.5% compared to Arm B. Our study design lacked the pill counts. Confirmation of drug compliance should be important. We only collected *EGFR* mutational status as driver mutations. As already discussed, male and ever smoker cases were more frequent in Arm A than in Arm B. For analyzing secondary endpoints RFS and OS, particularly in the subset analyses, the results should be interpreted carefully. Because of the small sample size, in multiple comparisons that may require adjustment of type 1 error, the current study does not have power for those secondary endpoints and any statement related to the significant/non-significant results would potentially be misleading.

In conclusion, although several adverse effects were less frequent in Arm A, both alternate-day and daily oral administration of S-1 were demonstrated to be feasible as adjuvant chemotherapy in elderly patients with stage IA to IIIA NSCLC. Although there is no standard care of postoperative adjuvant chemotherapy for elderly patients with completely resected NSCLC, future investigation should be focused on high-risk populations for recurrence.

## Supporting information

**S1 Checklist. CONSORT 2010 checklist of SLCG1201.**
(DOC)

**S1 Fig. Treatment schedule of S-1 as adjuvant chemotherapy for both arms.** Arm A: alternate-day administration, Arm B: daily administration.
(PDF)

**S2 Fig. Disease-specific (lung cancer-specific) survival of all the patients enrolled in this study.**
(PDF)

**S3 Fig.** **(A)** Recurrence-free survival (RFS) of the patients by completion of S-1 administration at 6 months. **(B)** Recurrence-free survival (RFS) of the patients by completion of S-1 administration at 6 months excluding the recurrent cases within 6 months (6 cases were excluded). **(C)** Overall survival (OS) of the patients by completion of S-1 administration at 6 months. **(D)** Overall survival (OS) of the patients by completion of S-1 administration at 6 months excluding the recurrent cases within 6 months (6 cases were excluded). **(E)** Recurrence-free survival (RFS) of the patients with the completion of S-1 administration at 6 months. **(F)** Overall survival (OS) of the patients with the completion of S-1 administration at 6 months. **(G)** Recurrence-free survival (RFS) of the patients without the completion of S-1 administration at 6 months. **(H)** Recurrence-free survival (RFS) of the patients without the completion of S-1 administration at 6 months excluding the recurrent cases within 6 months (6 cases were excluded). **(I)** Overall survival (OS) of the patients without the completion of S-1 administration at 6 months. **(J)** Overall survival (OS) of the patients without the completion of S-1 administration at 6 months excluding the recurrent cases within 6 months (6 cases were excluded). **(K)** Disease-specific survival (DSS) of the patients with the completion of S-1 administration at 6 months. **(L)** Disease-specific survival (DSS) of the patients without the completion of S-1 administration at 6 months. **(M)** Disease-specific survival (DSS) of the patients without the completion of S-1 administration at 6 months excluding the recurrent cases within 6 months (6 cases were excluded).
(PDF)

**S4 Fig.** **(A)** Recurrence-free survival (RFS) of the patients by pStage. **(B)** Overall survival (OS) of the patients by pStage. **(C)** Recurrence-free survival (RFS) of the patients with pStage IA

(T1bN0M0)/IB. **(D)** Recurrence-free survival (RFS) of the patients with pStage IB. **(E)** Overall survival (OS) of the patients with pStage IA (T1bN0M0)/IB. **(F)** Overall survival (OS) of the patients with pStage IB. **(G)** Recurrence-free survival (RFS) of the patients with pStage IIA/IIB. **(H)** Overall survival (OS) of the patients with pStage IIA/IIB. **(I)** Recurrence-free survival (RFS) of the patients with pStage IIIA. **(J)** Overall survival (OS) of the patients with pStage IIIA.
(PDF)

**S5 Fig. (A)** Recurrence-free survival (RFS) of the patients by *EGFR* mutational status. **(B)** Overall survival (OS) of the patients by *EGFR* mutational status.
(PDF)

**S6 Fig. (A)** Recurrence-free survival (RFS) of the patients by smoking status. **(B)** Overall survival (OS) of the patients by smoking status.
(PDF)

**S7 Fig. (A)** Recurrence-free survival (RFS) of the patients by sex. **(B)** Overall survival (OS) of the patients by sex.
(PDF)

**S1 Table. The exclusion criteria.**
(XLSX)

**S2 Table. The criteria for discontinuation and restart of S-1 administration.**
(XLSX)

**S3 Table. Criteria for dose reduction of S-1.**
(XLSX)

**S4 Table. Method for dose reduction of S-1.**
(XLSX)

**S5 Table. The criteria for cessation of the protocol treatment.**
(XLSX)

**S6 Table. Follow-up assessment.**
(XLSX)

**S7 Table. Enrolled and candidate cases for SLCG1201 study in each participating institution.**
(XLSX)

**S8 Table. Causes of death (n = 101).**
(XLSX)

**S9 Table. Actual total administered dose of S-1 in patients with (n = 64) or without (n = 33) treatment completion at 6 months (n = 97).**
(XLSX)

**S10 Table. The ratio of male to female per pStage (n = 101).**
(XLSX)

**S11 Table. Smoking status per pStage (n = 101).**
(XLSX)

**S1 Data. Detailed patient data.**
(XLSX)

**S2 Data. Detailed patient data.**
(XLSX)

**S3 Data. Detailed information of S-1 administration in each case.**
(XLSX)

**S4 Data. Detailed information of S-1 toxicity in each case.**
(XLSX)

**S1 File. English version of study protocol.**
(PDF)

**S2 File. English version of study protocol exhibit.**
(PDF)

**S3 File. Japanese version of study protocol.**
(PDF)

**S4 File. Japanese version of study protocol exhibit.**
(PDF)

## Acknowledgments

We thank Ms. Yumi Miyashita, the head Clinical Research Coordinator dispatched from a non-profit organization Epidemiological & Clinical Research Information Network (ECRIN), Kyoto, Japan, for data collection and management.

All the authors are the members of Setouchi Lung Cancer Group.

## Author Contributions

**Conceptualization:** Junichi Soh, Katsuyuki Hotta, Hiroshige Yoshioka, Hiroshi Date, Shinichi Toyooka.

**Data curation:** Hiromasa Yamamoto, Junichi Soh, Norihito Okumura, Hiroyuki Suzuki, Masao Nakata, Toshiya Fujiwara, Kenichi Gemba, Isao Sano, Takuji Fujinaga, Masafumi Kataoka, Yasuhiro Terazaki, Nobukazu Fujimoto, Kazuhiko Kataoka, Shinji Kosaka, Motohiro Yamashita, Hidetoshi Inokawa, Masaaki Inoue, Hiroshige Nakamura, Yoshinori Yamashita.

**Formal analysis:** Hiromasa Yamamoto, Junichi Soh, Norihito Okumura, Satoshi Morita, Keitaro Matsuo, Junichi Sakamoto, Hiroshi Date, Shinichi Toyooka.

**Investigation:** Katsuyuki Hotta, Hiroshige Yoshioka, Satoshi Morita, Keitaro Matsuo, Junichi Sakamoto, Shinichi Toyooka.

**Supervision:** Hiroshi Date, Shinichi Toyooka.

**Writing – original draft:** Hiromasa Yamamoto.

**Writing – review & editing:** Hiromasa Yamamoto, Junichi Soh, Norihito Okumura, Shinichi Toyooka.

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
