## [Decision Letter · Decision Letter 0]

26 Oct 2022

PONE-D-22-18798Randomized phase II study of daily versus alternate-day administrations of S-1 for the elderly patients with completely resected pathological stage IA (tumor diameter > 2 cm) - IIIA of non-small cell lung cancer: Setouchi Lung Cancer Group Study 1201PLOS ONE

Dear Dr. Yamamoto,

Thank you for submitting your manuscript to PLOS ONE. After careful consideration, we feel that it has merit but does not fully meet PLOS ONE’s publication criteria as it currently stands. Therefore, we invite you to submit a revised version of the manuscript that addresses the points raised during the review process. Your manuscript has been assessed by two peer-reviewers and their reports are appended below.  The reviewers comment that your manuscript would benefit from additional information and clarification. For example, the CONSORT diagram could be expanded to include information on eligibility before enrolment, and more details the study should expand on the selection of the choice between arm A and B. In addition, the reviewers comment that the statistics should be reported more carefully to ensure the outcome is not overstated and/or misleading.  Could you please carefully revise the manuscript to address all comments raised?

We look forward to receiving your revised manuscript.

Kind regards,

Maria Elisabeth Johanna Zalm, Ph.D

Editorial Office

PLOS ONE

Journal Requirements:

This work was supported by a non-profit organization Epidemiological & Clinical Research Information Network (ECRIN), Kyoto, Japan. 

I have read the journal's policy and the authors of this manuscript have the following competing interests:HY1 received honoraria from Taiho Pharmaceutical outside the work. JS1 received honoraria from Johnson and Johnson and Intuitive outside the work. HN received grants from Taiho Pharmaceutical outside the work. KH received honoraria from Pfizer, AstraZeneca, Chugai Pharmaceutical, Eli Lilly, Takeda Pharmaceutical, MSD, BMS, Ono Pharmaceutical, Taiho Pharmaceutical and Boehringer-Ingelheim outside the work, and grants from MSD, AstraZeneca, Chugai Pharmaceutical, Eli Lilly, BMS and Abbvie outside the work. HY2 received consulting fees from Delta-Fly Pharma and honoraria for lectures from AstraZeneca, Boehringer-Ingelheim, Chugai Pharmaceutical, Eli Lilly, Taiho Pharmaceutical, Ono Pharmaceutical, MSD, Novartis, BMS, Pfizer, Daiichi-Sankyo, Kyowa Kirin, Takeda Pharmaceutical, Nippon Kayaku and Otsuka Pharmaceutical outside the work. SM received honoraria for lectures from Taiho Pharmaceutical outside the work. HD received grants from Taiho Pharmaceutical outside the work. ST received grants from Chugai Pharmaceutical, Taiho Pharmaceutical, Eli Lilly, PFDeNA, MSD, AstraZeneca and a non-profit organization West Japan Oncology Group (WJOG) (supported by AstraZeneca for the operating expense of the clinical trial) outside the work. All other authors declared no conflicts of interest regarding this study. 

Reviewers' comments:

Reviewer's Responses to Questions

**Comments to the Author**

1. Is the manuscript technically sound, and do the data support the conclusions?

Reviewer #1: Partly

Reviewer #2: Yes

2. Has the statistical analysis been performed appropriately and rigorously? 

Reviewer #1: No

Reviewer #2: Yes

3. Have the authors made all data underlying the findings in their manuscript fully available?

Reviewer #1: Yes

Reviewer #2: Yes

4. Is the manuscript presented in an intelligible fashion and written in standard English?

Reviewer #1: Yes

Reviewer #2: Yes

5. Review Comments to the Author

Reviewer #1: Comments:

- The design: Why choosing feasibility based on the compliance as a primary endpoint for the study? How is compliance to treatment correlated to the harder endpoints such as OS or Disease free survival. Were there any data to support the surrogacy? Understand that the current endpoint can be observed earlier (at 6 months) but the study is conducted for much longer time and there is a good opportunity to observe harder endpoint such as Recurrence free or progression survival rate.

- Treatment completion was 69.4% in Arm A and 64.6% in Arm B yet the study Conclusion: Alternate-day oral administration of S-1 was demonstrated to be feasible with low toxicity in elderly patients with completely resected NSCLC solely focused on Arm A. Why was treatment B not mentioned to be feasible? Since both treatment arms passed the criteria to be feasible, was there any rule for choosing the better treatment Arm applied?

- In the current study there seems to be a trend that arm B is slightly better in term of OS and recurrence free survival. If phase II is warrantied, how will this be taking into consideration given the fact that feasibility show that arm A has slightly better compliance?

- Lines 395-398 “However, its drug discontinuation free survival rate, which is similar to the treatment completion Yamamoto et al. SLCG1201 19 rate at 1 year, was 52.7 %, suggesting that the compliance is not satisfactory despite no treatment-related death and low rate of any severe adverse events.” Question: How is this with respect to your feasibility criteria(hypothesis). Is not Arm B passing the treatment completion rate according to the hypothesis in the protocol?

- Anti DP-L1 therapies are introduced for the study population/in the market during the study conduct. Were any of the patients received anti-PDl1 in this study? Was there any data collected related to the use of anti PD-L1 therapy(ies)?

- With the existence of new anti PD-1/L1 in this patient population, how do you see the possibility of running phase III study based on the current study?

- Was subset analysis of survival in page 357 pre-planned in the protocol or in SAP? If not why was this analysis chosen?

- The statement about not significantly difference in this article, when analyzing secondary endpoints and particularly when in the subgroup analyses should be interpreted carefully. In addition to small sample size, multiple comparisons that may require adjustment of type 1 error, the study was not powered for those secondary endpoints and any statement related significant/non-significant would potentially be misleading.

Minor:

- Relapse free and recurrence free survival used in this article while presumably means the same thing. Please confirm and if so, please choose 1 term consistently throughout the article

- Reference on randomized phase II selection design, please use the original paper of Simon et al.(1985 *)

- P-values: when less than 0.001 should simply be stated (p < 0.001)

- Toxicity: as this a phase II study, would suggest that toxicity table should be included in the main body (not as supplementary)

*R Simon, R E Wittes, S S Ellenberg : Randomized phase II clinical trials

Comparative Study Cancer Treat Rep. 1985 Dec;69(12):1375-81.

Reviewer #2: Authors present results from a Phase II clinical trial of 1-year treatment with S-1 in elderly patients with completely resected pathological stage IA to IIIA non-small cell lung cancer (NSCLC), comparing two different cycles of treatment: 1) alternate-day oral administration of S-1 for 4 days a week; and 2) daily administration of S-1 for 2 weeks followed by a 1 week-rest (each cycle is repeated for the entire year.) Authors report on feasibility, safety and preliminary assessments of efficacy. The manuscript will be strengthened if the authors consider the following points.

1. Figure 1 is a CONSORT diagram. Ideally it would include a box before the 1st box, which shows how many people were considered for eligibility before enrollment. It seems highly unlikely that authors only considered 101 individuals for the study, and they all met inclusion/exclusion criteria.

2. A primary outcome being assessed is RDI which is the ratio of actual total administration dose and planned total administration dose. However, in the discussion, they mention a limitation of no pill counts. What is then meant by "actual total administration dose"?

Minor points:

1. I was confused by S1 Fig until I read the protocol, since I did not see mention in the text of the manuscript that S-1 was administered twice a day. Authors should state this in the main text, since that seems to be an important detail about the administration of the treatment. This may not be known by all readers.

2. Authors should specifically state in the statistical analysis section that they are performing "intent to treat" analyses for the survival outcomes.

3. In general, I would recommend rounding p-values - 3 digits beyond the decimal are a bit excessive, especially given the sample size. 1 or 2 non-zero digits beyond the decimal should be sufficient.

4. lines 331-332: authors already mentioned the lack of treatment related deaths in lines 327-328.

5. line 345: I suspect "monitored for at least 5 years from the registration" needs some clarification. The range of follow-up time was 6.2 to 103.9 months, so not everyone was followed for 5 years.

6. In the "Subset analysis of survival" section, authors should clarify which figures actually correspond to their statements. For example, lines 357-359 refer the reader to S3 Fig A-D, yet the sentence really only talks about S3 Fig B & D. Authors do not seem to mention the results of S3 Fig A & C.

6. PLOS authors have the option to publish the peer review history of their article (what does this mean?). If published, this will include your full peer review and any attached files.

Reviewer #1: No

Reviewer #2: No

---

## [Author Response · Author response to Decision Letter 0]

17 Dec 2022

We responded to the comments by reviewers and editor in attached files (Yamamoto et al PONE-D-22-18798 Responses to the Reviewers, Yamamoto et al Responses to Journal Requirements).

---

## [Decision Letter · Decision Letter 1]

17 Jan 2023

PONE-D-22-18798R1Randomized phase II study of daily versus alternate-day administrations of S-1 for the elderly patients with completely resected pathological stage IA (tumor diameter > 2 cm) - IIIA of non-small cell lung cancer: Setouchi Lung Cancer Group Study 1201PLOS ONE

Dear Dr. Yamamoto,

Thank you for submitting your manuscript to PLOS ONE. After careful consideration, we feel that it has merit but does not fully meet PLOS ONE’s publication criteria as it currently stands. Therefore, we invite you to submit a revised version of the manuscript that addresses the points raised during the review process.

The manuscript has been evaluated by two reviewers, and their comments are available below. Reviewer 1 has raised major concerns. They request improvements to the reporting of methodological aspects of the study, for example, appropriate reporting of the primary outcome of the trial (the completion rate at 6 months).

Could you please carefully revise the manuscript to address all comments raised?

We look forward to receiving your revised manuscript.

Kind regards,

Caroline Bull

Staff Editor

PLOS ONE

Journal Requirements:

Additional Editor Comments (if provided):

Reviewers' comments:

Reviewer's Responses to Questions

**Comments to the Author**

1. If the authors have adequately addressed your comments raised in a previous round of review and you feel that this manuscript is now acceptable for publication, you may indicate that here to bypass the “Comments to the Author” section, enter your conflict of interest statement in the “Confidential to Editor” section, and submit your "Accept" recommendation.

Reviewer #1: (No Response)

Reviewer #2: All comments have been addressed

2. Is the manuscript technically sound, and do the data support the conclusions?

Reviewer #1: Partly

Reviewer #2: (No Response)

3. Has the statistical analysis been performed appropriately and rigorously? 

Reviewer #1: No

Reviewer #2: (No Response)

4. Have the authors made all data underlying the findings in their manuscript fully available?

Reviewer #1: Yes

Reviewer #2: (No Response)

5. Is the manuscript presented in an intelligible fashion and written in standard English?

Reviewer #1: No

Reviewer #2: (No Response)

6. Review Comments to the Author

Reviewer #1: Thanks for your response. Concerning the following:

- Treatment completion was 69.4% in Arm A and 64.6% in Arm B yet the study conclusion: Alternate-day oral administration of S-1 was demonstrated to be feasible with low toxicity in elderly patients with completely resected NSCLC solely focused on Arm A. Why was treatment B not mentioned to be feasible? Since both treatment arms passed the criteria to be feasible, was there any rule for choosing the better treatment Arm applied?

From your answer:

“ …

As for the treatment completion rate, Table 3 shows that the completion rate at 12 months in Arm A (63.3%) is more than 15% higher than that in Arm B (45.8%).

“

This question is not answered appropriately. The primary endpoint (used to power the study) is the completion rate at 6 months and what you quoted here is the treatment completion rate at 12 months. You need to have your decision concerning feasibility of the study be based on your primary endpoint. The original question still needs to be addressed.

In addition, you should stick with the below rule, keeping in mind your primary endpoint as stated above. In this case treatment arm A and B both are successful wrt the primary endpoint and the difference between the 2 is much less than 15%.

“If the completion rate in one group is greater than 40% and the difference between one group and the other is within 15%, the two groups will not be directly compared based on retention rate numbers alone, but rather on toxicity, quality of life, convenience, cost, retention rate after 6 months, recurrence-free survival, and survival rate to determine which is more suitable as S-1 therapy in future Phase III trials.”

The following sentence in the discussion and throughout the manuscript should be corrected

As shown in Table 2, RDI of S-1 at 12 months is significantly better in Arm A than in Arm B. As for the treatment completion rate, Table 3 shows that the completion rate at 12 months in Arm A (63.3%) is more than 15% higher than that in Arm B (45.8%) ….

Reviewer #2: (No Response)

7. PLOS authors have the option to publish the peer review history of their article (what does this mean?). If published, this will include your full peer review and any attached files.

Reviewer #1: No

Reviewer #2: No

---

## [Author Response · Author response to Decision Letter 1]

31 Jan 2023

We responded to the comments from the Reviewer #1 and revised the manuscript as described in the attached files.

---

## [Decision Letter · Decision Letter 2]

28 Feb 2023

PONE-D-22-18798R2Randomized phase II study of daily versus alternate-day administrations of S-1 for the elderly patients with completely resected pathological stage IA (tumor diameter > 2 cm) - IIIA of non-small cell lung cancer: Setouchi Lung Cancer Group Study 1201PLOS ONE

Dear Dr. Yamamoto,

Thank you for submitting your manuscript to PLOS ONE. After careful consideration, we feel that it has merit but will fully meet PLOS ONE publication requirements after you address the points raised during this final review process.

We look forward to receiving your revised manuscript.

Kind regards,

Darren Wan-Teck Lim

Academic Editor

PLOS ONE

Journal Requirements:

Reviewers' comments:

Reviewer's Responses to Questions

**Comments to the Author**

1. If the authors have adequately addressed your comments raised in a previous round of review and you feel that this manuscript is now acceptable for publication, you may indicate that here to bypass the “Comments to the Author” section, enter your conflict of interest statement in the “Confidential to Editor” section, and submit your "Accept" recommendation.

Reviewer #1: All comments have been addressed

Reviewer #2: All comments have been addressed

Reviewer #3: All comments have been addressed

Reviewer #4: All comments have been addressed

Reviewer #5: (No Response)

2. Is the manuscript technically sound, and do the data support the conclusions?

Reviewer #1: (No Response)

Reviewer #2: (No Response)

Reviewer #3: Yes

Reviewer #4: Yes

Reviewer #5: Yes

3. Has the statistical analysis been performed appropriately and rigorously? 

Reviewer #1: (No Response)

Reviewer #2: (No Response)

Reviewer #3: Yes

Reviewer #4: Yes

Reviewer #5: Yes

4. Have the authors made all data underlying the findings in their manuscript fully available?

Reviewer #1: (No Response)

Reviewer #2: (No Response)

Reviewer #3: Yes

Reviewer #4: Yes

Reviewer #5: Yes

5. Is the manuscript presented in an intelligible fashion and written in standard English?

Reviewer #1: (No Response)

Reviewer #2: (No Response)

Reviewer #3: Yes

Reviewer #4: Yes

Reviewer #5: Yes

6. Review Comments to the Author

Reviewer #1: (No Response)

Reviewer #2: (No Response)

Reviewer #3: The study compared two different dose of S-1 for elderly NSCLC stage IA patients after surgery. The statistical design was acceptable, and the data were analyzed properly. The study showed that alternative schedule of S-1 may have similar efficacy to standard dose of S-1, with more tolerable toxicity. The result is also important for other indications of S-1, especially for elderly patients.

One question for the author: since UFUR is the only drug approved for the indication of adjuvant therapy for NSCLC, why did the author not consider using UFUR in two different dosing schedule?

Reviewer #4: (No Response)

Reviewer #5: 1. "the threshold 6-month treatment completion rate for the current protocol in both groups was 40%' - seems to be an extremely low threshold for adjuvant treatment set. Shouldn't a higher threshold be more clinically meaningful? eg 50% at least.

2. 1) If the 6-month treatment completion rate is 40% or less in both groups, the protocol treatments for both groups is considered not promising for postoperative adjuvant chemotherapy - why is 6-month completion rate used as primary endpoint, when the adjuvant treatment is for a duration of one year?

3. EGFR mutation was detected in 31 (30.7%) patients - looks like a big confounding factor, were these patients treated with osimertinib?

4. Median actual total administered dose of S-1 was 17340 mg 328 for Arm A and 17640 md (typo) for Arm B, respectively (p = 0.82)

5. To determine which is more suitable as S-1 therapy in future Phase III trials, the comparison between the two groups should not be performed with the treatment completion rate alone, but rather with toxicity, quality of life, convenience, cost, treatment completion rate after 6 months, RFS, and OS - given the "toxicities were generally well tolerated in both groups and there were no grade 4 adverse events", and numerically lower RFS (56.9% vs 65.7%) and OS (68.6% vs 82.0%) with the arm A, why did the authors come to a conclusion that arm A is 'more advantageous due to lower toxicity'? The numerical difference in survival rate is huge (and sample not adequate to get a low p value).

Another conclusion should be derived, which is "the experimental regime (arm A) has not been demonstrated to be a better alternative to the standard daily administration of TS1, and daily dosing remains the standard of care."

7. PLOS authors have the option to publish the peer review history of their article (what does this mean?). If published, this will include your full peer review and any attached files.

Reviewer #1: No

Reviewer #2: No

Reviewer #3: No

Reviewer #4: No

Reviewer #5: No

---

## [Author Response · Author response to Decision Letter 2]

9 Apr 2023

PONE-D-22-18798R2 

Yamamoto et al. 

Randomized phase II study of daily versus alternate-day administrations of S-1 for the elderly patients with completely resected pathological stage IA (tumor diameter > 2 cm) - IIIA of non-small cell lung cancer: Setouchi Lung Cancer Group Study 1201 

Responses to Journal Requirements 

Response. 

We reviewed our reference list and confirmed that it was complete and correct. We did not cite any papers that have been retracted. 

Responses to the Reviewers 

We thank the reviewers and the editor for their careful and conscientious reading of the manuscript and for their useful suggestions that have resulted in a much-improved manuscript. As indicated in the responses that follow, we have taken the comments and suggestions into account as much as possible in the revised version of our paper. The lines we indicate in the following comments are based on the revised manuscript with track changes. 

Comments by Reviewer #3 and our responses to each comment 

Reviewer #3: The study compared two different dose of S-1 for elderly NSCLC stage IA patients after surgery. The statistical design was acceptable, and the data were analyzed properly. The study showed that alternative schedule of S-1 may have similar efficacy to standard dose of S-1, with more tolerable toxicity. The result is also important for other indications of S-1, especially for elderly patients. 

One question for the author: since UFUR is the only drug approved for the indication of adjuvant therapy for NSCLC, why did the author not consider using UFUR in two different dosing schedule? 

Response. 

We are grateful for the comments from Reviewer #3. UFT has low toxicity compared to S-1. Daily administration of UFT has been already established and thus, we did not consider the study using UFT in two different dosing schedules. 

Comments by Reviewer #5 and our responses to each comment 

Reviewer #5: 

1. "the threshold 6-month treatment completion rate for the current protocol in both groups was 40%' - seems to be an extremely low threshold for adjuvant treatment set. Shouldn't a higher threshold be more clinically meaningful? eg 50% at least. 

Response. 

We are grateful for the comments from Reviewer #5. The objective of the current study is elderly patients, and we considered that the treatment completion rate of S-1 in elderly patients was lower than the patients younger than 75 years old. Although we set the treatment completion rate of S-1 as 60% in our previous study targeting the patients younger than 75 years old (SLCG1301, Ref. 26), we assumed that the threshold 6-month treatment completion rate for the current protocol in both groups was 40%. 

2. 1) If the 6-month treatment completion rate is 40% or less in both groups, the protocol treatments for both groups is considered not promising for postoperative adjuvant chemotherapy - why is 6-month completion rate used as primary endpoint, when the adjuvant treatment is for a duration of one year? 

Response. 

The objective of the current study is elderly patients, and we initially considered the possibility that only small number of cases continued the administration of S-1 for one year. Thus, we set the treatment completion rate at 6 months as the primary endpoint. 

3. EGFR mutation was detected in 31 (30.7%) patients - looks like a big confounding factor, were these patients treated with osimertinib? 

Response. 

EGFR mutational status was one of the adjustment factors for randomization and thus there is no difference in the frequency of EGFR mutations in both arms. We had 33 patients who had the recurrence in 101 enrolled patients. Among 33 patients, 10 patients had EGFR mutations and 2 patients received osimertinib. As for other EGFR-TKIs, 4 patients received gefitinib, 1 patient received afatinib, and 1 patient received erlotinib. 

4. Median actual total administered dose of S-1 was 17340 mg for Arm A and 17640 md (typo) for Arm B, respectively (p = 0.82) 

Response. 

This is just the description in “Feasibility” of “Results” of the main text, and there are no further comments or questions from Reviewer #5. 

5. To determine which is more suitable as S-1 therapy in future Phase III trials, the comparison between the two groups should not be performed with the treatment completion rate alone, but rather with toxicity, quality of life, convenience, cost, treatment completion rate after 6 months, RFS, and OS - given the "toxicities were generally well tolerated in both groups and there were no grade 4 adverse events", and numerically lower RFS (56.9% vs 65.7%) and OS (68.6% vs 82.0%) with the arm A, why did the authors come to a conclusion that arm A is 'more advantageous due to lower toxicity'? The numerical difference in survival rate is huge (and sample not adequate to get a low p value). 

Another conclusion should be derived, which is "the experimental regime (arm A) has not been demonstrated to be a better alternative to the standard daily administration of TS1, and daily dosing remains the standard of care." 

Response. 

Regarding OS and RFS, there is a trend of unfavorable prognosis in Arm A. However, there is no statistical difference between them. As described in Discussion, disease-specific survival showed no difference in both arms, and Arm A had male and ever smoker cases more frequently compared to Arm B. They may have affected the trend of unfavorable prognosis in Arm A. 

Although toxicities were generally well tolerated in both groups, anorexia, skin symptoms, and lacrimation, which are the typical adverse effects that can affect the patients’ quality of life, were significantly more frequent in Arm B compared with Arm A.

As described in Discussion, there is no standard care of postoperative adjuvant chemotherapy for elderly patients with completely resected NSCLC. Thus, “daily dosing remains the standard of care” raised by Reviewer #5 as another conclusion, is not appropriate. 

Considering above, we deleted the phrase “although alternate-day oral administration of S-1 was advantageous for better treatment completion at 12 months and low toxicity” and put the phrase of “Although several adverse effects were less frequent in Arm A,” and the following are the final version of Conclusion in Abstract and Discussion in the main text. 

In Abstract (Lines 76-80): 

Conclusion: Although several adverse effects were less frequent in Arm A, both alternate-day and daily oral administrations of S-1 were demonstrated to be feasible in elderly patients with completely resected NSCLC. 

In Discussion (Lines 493-499): 

In conclusion, although several adverse effects were less frequent in Arm A, both alternate-day and daily oral administration of S-1 were demonstrated to be feasible as adjuvant chemotherapy in elderly patients with stage IA to IIIA NSCLC. Although there is no standard care of postoperative adjuvant chemotherapy for elderly patients with completely resected NSCLC, future investigation should be focused on high-risk populations for recurrence.

---

## [Editor Report · Decision Letter 3]

19 Apr 2023

Randomized phase II study of daily versus alternate-day administrations of S-1 for the elderly patients with completely resected pathological stage IA (tumor diameter > 2 cm) - IIIA of non-small cell lung cancer: Setouchi Lung Cancer Group Study 1201

PONE-D-22-18798R3

Dear Dr. Yamamoto,

We’re pleased to inform you that your manuscript has been judged scientifically suitable for publication and will be formally accepted for publication once it meets all outstanding technical requirements.

Kind regards,

Darren Wan-Teck Lim

Academic Editor

PLOS ONE
---

## [Editor Report · Acceptance letter]

10 May 2023

PONE-D-22-18798R3 

Randomized phase II study of daily versus alternate-day administrations of S-1 for the elderly patients with completely resected pathological stage IA (tumor diameter > 2 cm) - IIIA of non-small cell lung cancer: Setouchi Lung Cancer Group Study 1201 

Dear Dr. Yamamoto:

I'm pleased to inform you that your manuscript has been deemed suitable for publication in PLOS ONE. Congratulations! Your manuscript is now with our production department. 

Kind regards, 

on behalf of

Dr. Darren Wan-Teck Lim 

Academic Editor

PLOS ONE